# Health-related behavioral changes during the COVID-19 pandemic. A comparison between cohorts of French and Italian university students

Ivana Matteucci[1]*, Mario Corsi[1], Rémy Hurdiel[2], Thierry Pezé[2], Philippe Masson[3], Alessandro Porrovecchio[2]

**1** Department of Communication Sciences, International Studies and Humanities, University of Urbino Carlo Bo, Urbino, Italy, **2** ULR 7369 - URePSSS Unité de Recherche Pluridisciplinaire Sport Santé Société, Univ. Littoral Côte d'Opale, Univ. Lille, Univ. Artois, Dunkerque, France, **3** ULR 7369 - URePSSS - Unité de Recherche Pluridisciplinaire Sport Santé Société, Univ. Lille, Univ. Artois, Univ. Littoral Côte d'Opale, Lille, France

☯ These authors contributed equally to this work.
* ivana.matteucci@uniurb.it

**Data Availability Statement:** All relevant data is available here: https://doi.org/10.5064/F6472XEL

## Abstract

This cross-sectional observational study compares the health behaviors of university students in France and Italy, examining how their choices and lifestyles were affected by the COVID-19 pandemic with the aim of contributing to the development of adequate public health and higher education institutions interventions. The French cohort was investigated between January and February 2022, while the Italian cohort was examined between March and April 2022. In both contexts, data were collected through web surveys using institutional directories of university degree programs. Data were collected using standardized tools, validated and recovered in full or partial form. The tool used consisted of three specific sections (general experience, eating habits, physical activity), to which a fourth, dedicated to describing the sociographic picture of the respondents, was added. It was found that the pandemic mainly affected the mental health and sense of well-being of young people in both countries. The pandemic altered dietary habits (41.8% of the French subjects and 38.3% of the Italians declared an increasing of their food intake), alcohol consumption (9.0% of the Italian respondents and 4.0% of the French respondents reported an increased alcohol consumption), propensity to smoke (among the French 85.3% subjects remained non-smokers versus 65.3% of the Italian subjects), sleep quality (25.7% of Italian students 16.6% of French students experienced a decline in the quality of their sleep), and physical activity levels (the percentage of physically active French subjects rose to 72.4%, whereas among Italian students, it dropped to 68,4%). The results emphasize the need for the implementation of relational and psychological interventions, even digital, to face the consequences of social isolation and negative changes in everyday behaviors due to the restrictions during the COVID-19 pandemic.

**Funding:** The source of funding that has supported the work is University of Urbino Carlo Bo - Department of Communication Sciences, International Studies and Humanities, Via Saffi 2, 61029 Urbino (PU). URL to sponsor' website: https://www.uniurb.it/. The funder had no role in study design, data collection and analysis, decision to publish, or preparation of the manuscript.

**Competing interests:** The authors have declared that no competing interests exist.

## Introduction

During the COVID-19 pandemic, the Italian government and local health authorities, imposed restrictions, which varied regionally according to the data and trends of each particular region, on travelling, also ordering the total closure of schools and universities, and from the outset, implementing a ban on outdoor sports and physical activities [1, 2]. The French government initially issued general recommendations to reduce unnecessary travel and social contacts, and subsequently implemented specific measures, including stay-at-home orders and limitations on work and educational activities [3].

In the literature, correlations between restrictive measures due to the COVID-19 pandemic and changes in behavior negatively impacting student health have been found [4–6], raising concerns that such new behaviors could become an integral part of the lifestyles of university students [7]. Essadek and Rabeyron [8] investigated the psychological and situational factors contributing to anxiety and depressive symptoms among French students and reported that symptoms of anxiety were particularly acute during the two lockdowns. According to Charbonnier et al. [9] these symptoms continued when the universities reopened. For several authors the main emergencies found in university students surveyed during the COVID-19 pandemic concern mental health risks and the possibility of experiencing negative emotions [10–19].

Du et al. [20] observed students who experienced a decline in the quality of their sleep during the COVID-19 pandemic had higher dietary risk scores than students who did not experience such a decline. Amatori et al. [21] investigated the effects of mood states and exercise on nutritional choices among Italian university students during the COVID-19 lockdown. They found that poorer mood states led to unhealthy dietary habits, which could thus be associated with negative moods. Goncalves et al. [22] reported that the pandemic appeared to have a positive effect on alcohol consumption. Moreover, sedentary levels were found to be higher during both lockdowns, and these sedentary behaviors tended to persist over time. Romero-Blanco et al. [23] observed that, during the lockdown, both weekly physical activity (especially among women) and weekly sitting time increased overall and by group.

Some authors [24] focused on psychological, emotional, and cognitive correlates of social confinement in a sample of college students, integrating qualitative and quantitative analyses to identify potential predictors of traumatic distress and low academic performance during the COVID -19 pandemic. They identified the problematic thinking style "all or nothing" as the strongest predictor of psychological distress, anxiety, depression, and posttraumatic symptoms. "Everything will be fine" was identified by the "optimistic style" inversely correlated with the psychopathological measures and concentration problems. A recent UK population survey [25] found that coping poorly, and pre-existing mental health conditions were significantly associated with suicidal thoughts and self-harm in young adults. An Australian study showed that resilient mindset mediated the relationship between coronavirus-related stress and depressive symptoms. Moreover, loneliness moderated the mediating effect of resilient mindset in the coronavirus stress and depressive symptoms association among college students [26]. A Chinese study [27] identified the trajectories and the predictors among sociodemographic and psychosocial variables at baseline of vicarious traumatization (VT) in Chinese college students during the COVID-19 pandemic. The authors declared that targeted psychological interventions are urgently needed for students vulnerable to VT. Another Italian study [28] investigated the impact of distance education (DE) on mental health, social cognition, and memory abilities in a sample of university students during the national COVID-19 lockdown in Italy and identified the predictors of academic performance. Half of the student

sample reported significant impairment in concentration and learning abilities during DE. Regarding psychological health 19.7%, 27.1%, and 23.6% of the sample reported mild, moderate, and severe depressive symptoms, respectively.

Because the student population is at greater risk of high stress and behavioral and psychological problems, and because many students have experienced significant destabilization of their daily lives, the present, predominantly observational study compares the health-related behaviors of university students in France and Italy, to recognize the impact that the pandemic has had on student's health and wellbeing. In particular, the comparison of the results relating to the two cohorts demonstrated that the health emergency may have played an active role in negatively impacting certain behaviors in both cohorts of university students. These results emphasize the need for the implementation of relational and psychological interventions, even digital, to help students to face the consequences of social isolation and negative changes in everyday behaviors due to the restrictions during the COVID-19 pandemic.

## Materials and methods

### Ethics

**Ethics statement.**   The study was approved by the "Commission Nationale de l'Informatique et des Libertés - CNIL" (The France Data Protection Authority).

Ethical review and approval is not required for the study on human participants in accordance with the local Italian legislation and institutional requirements. In fact, the data collection is completely anonymous and data are not described except in an aggregated way, this makes the data sources (students) unrecognizable. Furthermore, the questionnaire explicitly stated that the compilation automatically implied the release on the use of the data provided in the above modalities.

**Consent statement.**   Data collection was confidential, no personal information was collected (name, phone number, address, etc..): the purpose of the study was explained to the students and their written informed consent was obtained.

### Context of the investigation

Our investigation was partially based on the Universanté program, which had been underway since 2008 in the Opal Coast area of France. The aim of our study was to survey some aspects of life (secondary effects of the pandemic) related to university student cohorts. Specifically, using a comparative approach, we investigated possible differences between the two groups residing in different countries in terms of how those groups coexisted with the pandemic. The first group resided in the Opal Coast area, which is located in the Hauts de France region (France) (3,200 sq km, 800,000 inhab.). The second group resided in Central Italy (Marche region - 9,400 sq km, 1,500,000 inhab.). The target populations were university students enrolled in para-medical health degree programs (nursing, physiotherapy, etc.) and degree programs in Physical Education for Health and Prevention.

### Survey period

Information was collected in the post-acute phase of the third cycle of the pandemic. Specifically, the French cohort was investigated between January and February 2022, while the Italian cohort was examined between March and April 2022.

## Methodology

Based on investigations carried out in France, a suitable number of behaviors likely to be conditioned by the pandemic were selected from the literature. Data was collected for these behaviors using standardized tools, validated and recovered in full or partial form. The tool used in the Italian context which is part of a larger French data set, consisted of three specific sections, to which a fourth, dedicated to describing the sociographic picture of the respondents, was added. The first section examined the general experience of the students, also in comparative terms, before and during the pandemic, seeking to provide an initial picture of students' habitual daily behaviors. The second section focused on eating habits without neglecting possible deviations related to eating disorders (SCOFF questionnaire - Sick Control One Stone Fat Food) [29] and the possible use or abuse of alcohol and cigarettes. The third framework, based on a scale known in the literature as the IFIS questionnaire - International Fitness Scale [30] was used to assess the level of physical activity that characterized the subjects' daily lives. In both contexts, data were collected through web surveys using institutional directories of university degree programs. Students were invited to complete the survey by a message explaining the purposes of the initiative. This message was followed up, about one week later, by a reminder message, whose aim was to boost the participation rate.

The survey was organized by the authors of the research. The focus groups of students involved consisted of students enrolled in the same year of academic courses, the compilation of the questionnaire was solicited by peer tutors trained by the researchers. The tutors provided explanations of the research aims through an online forum and support for completing the questionnaire. The researchers built an Italian sample to be compared with a pre-existing French sample. To mitigate the influence of any covariates, an Italian cohort was selected referring to a demographic area comparable in size and population, with similar characteristics of university students. In particular, the latter condition led to recruit the entire student population with similar characteristics in the area. Data collection took place through web survey using institutional mailing lists. It was not possible to know how many e-mails sent were actually read, so probably the participation rate indicated is approximated by default. The students were contacted via the institutional mailing list with a message without a request of confirmation of reading (again for privacy), so the number of emails that were sent is known, but not the number of students who actually read them. For this reason, the indicated response rates represent a lower bound on the participation of student cohorts. That is, the real rate was equal to or higher than the one indicated.

## Results

### Participation

The Italian participation rate of 25.7% was much higher than the French rate, which was just over 10%. A total of 567 questionnaires were collected. Of those questionnaires, 70.5% were from the French cohort and the remaining 29.5% from the Italian one, percentages which reflected the general populations in the two contexts under investigation (approximately 73% and 27%).

### Characteristics of the respondents

The French cohort was predominantly female, 75.8%, whereas only 58.1% of the Italian cohort was female. The two anthropometric parameters, height and weight, did not differ substantially between the two groups, and any differences were largely attributable to the gender disparity between the two cohorts. These parameters were considered in terms of their

relationship to other individual variables. Turning our attention to 'health-care consumption', a telling behavior during a pandemic, we see that in the French context, a share equal to 15.5% reported having foregone some type of health-care service (general practitioner consultation, specialist visit, etc.), a percentage that this is clearly lower than the Italian percentage, which stands at 25.1% (Table 1).

## General experience

This initial overview allows us to draw a comparison, for some items, between 'normal' behaviors and those under pandemic-induced lockdowns. From the analysis of the data (Table 2), we can observe that 23.1% of the French students tested positive for COVID-19, while this value rose to 40.1% among the Italian students. Using an assessment scale from 1 to 10, respondents were asked to quantify their discomfort with the pandemic in generic terms of 'fear'. The synthesis of these assessments yielded very different results, showing that the French (average = 2.84, SD = 2.40) reacted better than their Italian peers (average = 4.08, SD = 2.40).

As regards food intake, 41.8% of the French subjects compared to 38.3% of the Italians, reported increasing their food intake during periods of confinement, a figure that is certainly

**Table 1. Sociographic characteristics of the respondents compared by participating cohort.** Percentages (modal values in bold) or absolute values.

| Item | France (n = 400) | Italy (n = 167) |
|---|---|---|
| **01- Field of study** | | |
| Degree programs in Physical education for health | 24.0 | **60.5** |
| Degree programs in Nursing | **72.5** | 7.2 |
| Other degree programs in non-medical health professions | 3.5 | 25.1 |
| Other types of degree programs | 0.0 | 7.2 |
| **02 - Body weight** | | |
| Average value | 67.5 | 66.1 |
| Standard deviation | 15.4 | 13.1 |
| **03 –Height** | | |
| Average value | 1.68 | 1.71 |
| Standard deviation | 0.09 | 0.10 |
| **04 –Age** | | |
| Average value | 21.3 | 24.1 |
| Standard deviation | 6.5 | 3.5 |
| **05 –Gender** | | |
| Male | 24.2 | 41.9 |
| Female | 75.8 | 58.1 |
| **06 –Deferment of medical needs in the last 12 months** | | |
| Yes | 15.5 | 25.1 |
| No | **84.5** | **74.9** |
| **07 - Affiliation with a general practitioner** | | |
| Yes | **94.9** | **82.6** |
| No | 5.1 | 17.4 |
| **08 - Annual medical visits** | | |
| Average value | 4.5 | 2.5 |
| Standard deviation | 4.3 | 2.1 |
| **09 - Consultation of medical specialists in the last 12 months** | | |
| Yes | **62.8** | **61.1** |
| No | 37.2 | 38.9 |

**Table 2. Overall experience of the respondents compared by participant cohort.** Percentages (modal in bold) or absolute values.

| Item | France (n = 400) | Italy (n = 167) |
|---|---|---|
| **A1- Tested positive for COVID** | | |
| Yes | 23.1 | 40.1 |
| No | **76.9** | **59.9** |
| **A2 - Persistence of post-illness symptoms** | | |
| Yes | 6.8 | 5.4 |
| No | **93.2** | **94.6** |
| **A3 - Self-assessment level of fear of the pandemic** | | |
| Average value | 2.84 | 4.08 |
| Standard deviation | 2.40 | 2.40 |
| **A4 - Changes in diet during lockdown** | | |
| Yes, I ate moderately more | **41.8** | 38.3 |
| Yes, I ate moderately less | 17.6 | 22.8 |
| No, I ate as before | 40.6 | **38.9** |
| **A5 - Changes in alcohol consumption during lockdown** | | |
| Increased | 4.0 | 9.0 |
| It remained the same | **55.0** | 24.0 |
| Decreased | 28.7 | **38.3** |
| I don't drink alcohol | 12.3 | 28.7 |
| **A6 - Change in smoking during lockdown** | | |
| I remained a non-smoker | **85.3** | **65.3** |
| I smoked more than before | 4.2 | 9.0 |
| I smoked as much as before | 6.1 | 8.4 |
| I smoked less than before | 1.7 | 8.4 |
| I stopped smoking | 2.2 | 8.4 |
| I started smoking again | 0.6 | 0.6 |
| **A7 –Physical activity before the lockdown** | | |
| Yes | **63.3** | **80.2** |
| No | 36.7 | 19.8 |
| **A8 –Physical activity during the lockdown** | | |
| No | 27.6 | 31.6 |
| Yes, with the same frequency as before | 21.5 | 16.7 |
| Yes, but less frequently than before | **30.4** | **41.2** |
| Yes, I started during the lockdown | 20.5 | 10.5 |
| **A9 - Places where physical activity is mainly done** | | |
| At home | **59.7** | **79.2** |
| Outdoor | 40.3 | 20.8 |
| **A10 - Regularity of sleep during lockdown** | | |
| I slept better than usual | 31.0 | 9.0 |
| I slept normally | **52.4** | **65.3** |
| I did not sleep as well as usual | 16.6 | 25.7 |
| **A11 - Regular sleep after lockdown** | | |
| I slept better than usual | 7.9 | 3.0 |
| I slept normally | **70.0** | **74.3** |
| I did not sleep as well as usual | 22.1 | 22.7 |

in line with the greater amount of time spent at home. On the other hand, regarding alcohol consumption, the Italian cohort, with 9.0% of respondents reporting increased alcohol consumption, clearly outpaced the French cohort, with only 4.0% of respondents reporting an increase. In neither of the two territorial contexts did we find a significant dependence of behaviors on gender.

We found a lower propensity to smoke among the French (85.3% of the French subjects were non-smokers versus 65.3% of the Italian subjects) and the difference between the two cohorts was maintained, though slightly reduced by the proportion of those who quit and those who started smoking again. Overall, smoking behaviors were very similar in both cohorts, although the situation of the French cohort was more critical. No significant differences were found between genders as regards subjects' propensity to smoke, even within the same territorial context. Before the implementation of restrictive measures, four out of five Italians (80.2%) were physically active compared to 63.3% of their French counterparts. Once these measures were in force, among the French subjects, the percentage of physically active subjects rose to 72.4% (summing the positive values), whereas among Italian students, it dropped to 68,4%.

The last two items in the questionnaire concern the quality of sleep of the subjects, reflecting their psychic well-being in the lockdown and post-lockdown periods. Thus, when stay-at-home orders were in place, half of the French subjects (52.4%) and two out of three of their Italian counterparts (65.3%) slept normally. On the other hand, the percentage of Italian students experiencing a decline in the quality of their sleep was higher, 25.7%, than it was among the French cohort, of whom only 16.6% reported a decline in the quality of their sleep. The inevitable recovery after restrictions had been lifted saw the percentages of subjects not experiencing sleep problems rise in both cohorts (France 70.0%, Italy 74.3%), although nonnegligible percentages of both cohorts, about one in five, continued to report some sleep problems.

## Diet

This section of the questionnaire aims to assess the kind of lifestyle subjects adopted during the pandemic. It is divided into three main sections related to diet, the relationship that subjects have with food, and alcohol consumption. This detailed section concerning eating habits (S1 Table) can also be readily analyzed to detect the presence of any constructs in the articulation of the items. In this regard, an analysis of the main components was carried out using an exploratory approach, the results of which were interpreted in a descriptive manner. The structures that emerge in the two territorial contexts are very similar. Regarding the Italian reality, which has a more intelligible structure, we see a clear identification for the five factors extracted. The first of these contains all the items related to the concomitance of food intake at home and the active presence of television. It would thus seem to be confirmed that television (with its associated sitting time) becomes the main 'interlocutor' of young people, a fact perhaps attributable to the limitations imposed by lockdowns. The second factor, with an equally clear structure, encompasses all the items related to time spent eating, confirming that the habit was independent of the time of day and therefore of the type of meal consumed. The third factor highlights the virtuous concomitance of the consumption of fruit and vegetables, while the fourth brings together the habit of dining out and intake of sweet drinks. Finally, the fifth and last factor associates the concomitance of snacking and the intake of dairy products, an indication of good dietary habits.

The five items in the next scale concerning the relationship that subjects have with food (Table 3) provided an opportunity to construct a sentinel event indicating potential eating

**Table 3. Dietary habits designed to highlight the onset or presence of possible disorders (SCOFF) among the respondents compared by participating cohort.** Percentages (Modal values in bold) or absolute values.

| Item | France (n = 400) | Italy (n = 167) |
|---|---|---|
| Yes | 5,5 | **74,3** |
| No | **94,5** | 25,7 |
| **B20 - Do you worry you have lost control over how much you eat?** | | |
| Yes | 27,6 | **56,9** |
| No | **72,4** | 43,1 |
| **B21 - Have you recently lost more than 6 kg in a three-month period?** | | |
| Yes | 15,4 | 14,4 |
| No | **84,6** | **85,6** |
| **B22 - Do you think you are fat when others say you are too thin?** | | |
| Yes | 30,5 | 36,5 |
| No | **69,5** | **63,5** |
| **B23 - Would you say food dominates your life?** | | |
| Yes | 25,1 | 22,2 |
| No | **74,9** | **77,8** |

disorders. The approach followed was that of the SCOFF eating disorders questionnaire, a well-known screening tool according to which subjects who report two or more problems among the five under consideration are likely to be referred for a more detailed assessment [29]. The Italian students followed diets more than their French counterparts, appeared to be more concerned about their weight and more prone to developing eating disorders. Indeed, members of the Italian cohort often reported 'Feeling disgusted for having eaten too much', 'worrying about loss of control over food consumption', 'considering themselves fat even though others say they are thin' According to the SCOFF questionnaire valuation, 29.4% of the French students and 65.9% of their Italian counterparts warranted further assessment. The results of the questionnaire, especially those regarding the Italian cohort, which appear particularly worrisome, seem to point to the mental and psychological distress generated by the pandemic.

High percentages of both cohorts reported having consumed alcohol in the last twelve months, whereas frequent use of alcohol was more common among the Italian students. With regard to gender differences, there were no significant discrepancies, although, in the case of the French cohort, there was a certain discrepancy between males and females in favor of the former. For both cohorts the reasons for alcohol consumption did not appear to be linked to 'feeling good' or 'fighting loneliness', although for the Italian sample there was considerable percentage of subjects who reported drinking 'sometimes' to feel good psychologically (24,1%) and when they were alone (23.3%). The majority (similar percentages) of subjects in both cohorts reported that they did not have problems with alcohol assumption (Table 4).

## Physical activity

This section of the questionnaire aimed to assess physical fitness and focused on issues that can be used to gage fitness levels. Six of the nine items in the framework are, in fact, part of the scale for measuring physical fitness (IFIS Questionnaire) [30]. Respondents were asked to rank on a scale their general physical fitness, ability to withstand prolonged exertion, muscle strength, speed in movements, agility and flexibility in physical exercises (S2 Table).

Assigning the scale levels to the same number of equally spaced numerical values thus allowed us, assuming the equal importance of the aspects, to obtain, by simple additive

**Table 4. Practices pointing to the onset or presence of possible alcohol abuse among the respondents compared by participating cohort.** Percentages (Modal values in bold) or absolute values.

| Item | France (n = 400) | Italy (n = 167) |
|---|---|---|
| **B24- Consumption of alcoholic beverages in the last 12 months** | | |
| Yes | **85,4** | **89,8** |
| No | 14,6 | 10,2 |
| **B25 - Possible frequency of alcohol consumption** | (n = 303) | (n = 148) |
| Once a month | 43,4 | 21,6 |
| 2 to 4 times a month | **43,8** | **58,8** |
| 2 to 3 times a week | 11,4 | 18,9 |
| At least 4 times a week | 1,4 | 0,7 |
| **B26 - Use of alcohol to improve your psychological state** | (n = 303) | (n = 148) |
| Never | **90,4** | **70,5** |
| Sometimes (<1 once a month) | 8,3 | 24,1 |
| Often (once a month) | 1,3 | 5,4 |
| **B27 - Solitary use of alcohol** | (n = 303) | (n = 148) |
| Never | **93,8** | **76,7** |
| Sometimes (<1 once a month) | 5,9 | 23,3 |
| Often (once a month) | 0,3 | 0,0 |
| **B28 - Recommended reduction in alcohol use by family members / acquaintances** | (n = 303) | (n = 148) |
| Never | **96,4** | **90,6** |
| Sometimes (<1 time per month) | 2,6 | 8,7 |
| Often (once a month) | 1,0 | 0,7 |
| **B29 - Problems with alcohol consumption** | (n = 303) | (n = 148) |
| Never | **94,4** | **85,2** |
| Sometimes (<1 time per month) | 5,0 | 14,1 |
| Often (once a month) | 0,6 | 0,7 |

aggregation, a summary value. In our case, the summary value, renormalized in the range of 0–1, saw the Italian cohort, with an average of 0.66 (SD = 0.19), outperform its French counterpart, which showed a value of 0.59 (SD = 0.19). As shown by the graph (Fig 1), which also breaks down the data by gender, the relative distributions show the presence of some downward outliers presumably attributable to subjects experiencing problems. The fact that the Italian cohort was found to be in better shape than its French counterpart during the pandemic is likely due to the presence of students from the Physical Education for Prevention and Health degree program in the Italian cohort. Focusing on gender, the apparent superior performance of males is statistically significant in both contexts, though it is more marked in the French cohort.

Among the many items used in the survey, some of them, in aggregate or single form, are indicative of more complex aspects or behaviors. Thus, it was possible to determine three 'literature' indicators: physical condition measurement through the construction of the BMI (Body Mass Index), the SCOFF indicator on the potential or possible presence of eating disorders, and the IFIS indicator of physical fitness. To these indicators we can add the single item relating to the subjects' following or having followed diets, which can be useful in completing the picture related to the subjects' psychological condition, habits and lifestyle.

Before continuing our analysis, let us examine the behavior of the BMI indicator (Fig 2). The data disaggregated also by gender, show the presence of many outliers, especially in the French context. The average values of the measure show a significant difference between the

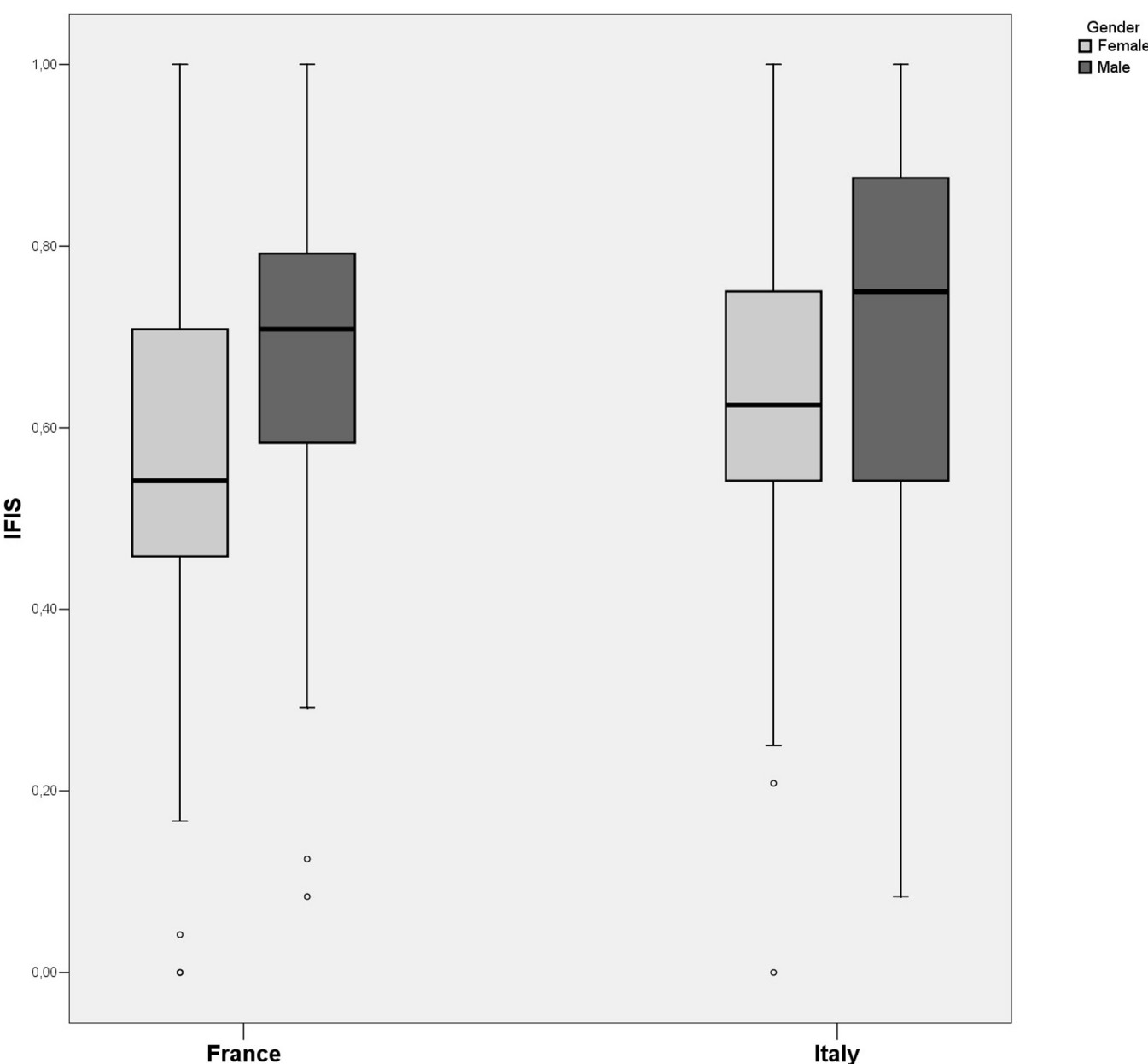

**Fig 1. Box-Plot representations of the IFIS measure regarding the physical fitness of the respondents compared by participant cohort and gender.**

territorial contexts, with the French students (M = 23.9, DS = 4.91) showing higher values than their Italian counterparts (M = 22.4, DS = 2.81). Again, the presence in the Italian cohort of students from the Physical Education for Health and Prevention degree program could account for the difference between the two groups. If we then look at gender, a significant difference was only found in the Italian context.

Returning to our global analysis, it is natural to ask ourselves if there are mutual relationships between the proposed synthesis measures that could be a consequence of causal relationships between the behaviors they describe. Starting from respondents who reported having resorted to diets, we can observe how this practice is positively correlated with BMI (r = 0.44, p <0.01 in the French cohort and r = 0.30, p <0.01 in the Italian one) and with the indicator for

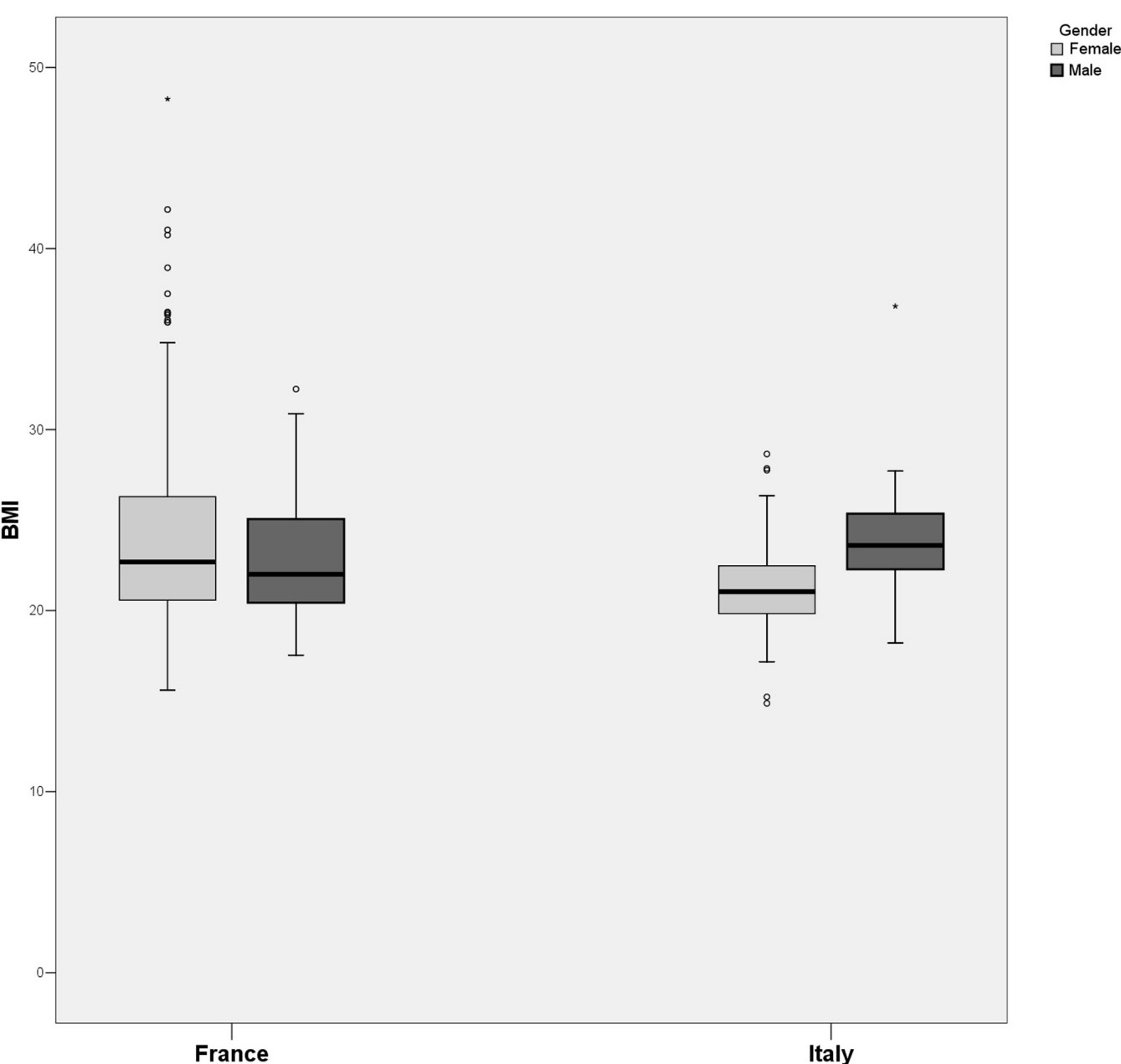

**Fig 2. Box-Plot representations of the BMI measurement of the respondents compared by participant cohort and gender.**

the possible presence of eating disorders (Sick Control One Stone Fat Food SCOFF) (r = 0.35, p <0.01 in the French cohort and r = 0.19, p <0.05 in the Italian one). Still considering respondents who reported having followed diets, the relationship changes, losing significance in the Italian case, when the comparison is made using the international fitness scale index (IFIS) (r = -0.17, p <0.01 in the French cohort and r = -0.12, n. s. in the Italian one). Hence, in view of all of these results, it would seem that the ambiguous presence of an active control over food intake may, on the one hand, respond to the need to lose weight by doing physical activity but, on the other hand, may also point to the presence of a possible eating disorder.

Continuing to examine the subjects' physical fitness levels using BMI measurement, we found that it was related to the IFIS physical fitness measure. Among the French students this relationship was found to be significant, with high BMI values associated with low levels of physical fitness (IFIS) e (r = -0.32, p <0.01), while among the Italian students no such relationship was found. In Italian students, therefore, there was no demonstrable correlation between following diets and high body mass on the one hand, and limited physical fitness on the other. This apparently contradictory finding could be explained by the fact that high BMI values may also be found in subjects with good level of physical fitness (IFIS), a characteristic that may be unique to students from the Physical Education for Health and Prevention degree program who made up most of the Italian cohort.

The same ambiguous correlation was found when relating BMI to the SCOFF indicator of eating disorders. Once again in the French cohort, a significant relationship was found with r = 0.30, p <0.01, showing how eating disorders are often the result of a physical appearance that is not aligned with current health and aesthetic models. As in the previous case, no such relationship was found among the Italian students confirming a sense of distress not directly attributable to body weight but possibly attributable to the 'burden' imposed by the pandemic conditions.

## Discussion

The results of the study suggest that the health emergency may have played an active role in negatively impacting certain behaviors in both cohorts of university students. For example, we observed a reduction in the quality of sleep (mainly in the Italian sample: +9.1%), increased consumption of alcohol (mainly in the Italian sample: +5.0%) and cigarette smoking (mainly in the French sample: +20.0%). These results are consistent with what has been reported in the literature, where some studies have observed an increase in tobacco and alcohol use among university students in response to pandemic-related stress [6, 14, 31–34].

The three literature indicators constructed with BMI, IFIS and SCOFF data show relationships between them that are not strong but often statistically significant. The important finding that emerges is that the pandemic seemed to have a greater impact on the Italian students. This finding might be related to the different evolution of the COVID-19 pandemic in Italy, where stricter lockdowns and extended distance learning were imposed by government authorities. These restrictions were more severe than those imposed by French authorities. The comparison showed both similarities and differences between the two cohorts. The differences could stem from the different ways of managing the pandemic by the governments and public health authorities in the two countries, with a consequent different impact on students' lifestyles. Nevertheless, the present analysis doesn't isolate "social restriction" as an independent causal factor that permit causal inference; in addiction, hypothesis regarding other possible causes are included.

The problems resulting from the pandemic affecting young people in particular concern their altered lifestyles, making them more prone to health problems than they were before the pandemic. Some researchers have raised the question of why and how to deal with the issue of assuming incorrect lifestyles associated with lockdowns in the short or medium term. Indeed, the unknown impacts on the mental health and well-being, fitness and eating behaviors of young people emerge as major issues of concern, not least because they may persist even after lockdowns have been lifted [35]. Overall, considering the young university student population, the emerging risk justifies proposals and raises the need by public health and higher education institutions calling for specific adaptations and types of activities designed to ensure health maintenance by balancing physiological aspects, physical fitness, and in particular,

psychological and socio-cultural factors, to prevent new unhealthy habits or routines from taking root in young people in the wake of the COVID-19 pandemic.

In particular, the study results emphasize the need for the implementation of social and psychological interventions, even digital, to face the consequences of social isolation and negative changes in everyday behaviors due to the restrictions during the COVID-19 pandemic. The universities whose students have been recruited regularly activate face-to-face listening and counseling services to help students to cope with emotional distress in academic tasks, but they did not activate similar services online during the pandemic. Scientific leterature provides evidence on the beneficial effects of digital mental health interventions for university students. Hovewer, due to the variation in study settings and inconsistencies in reporting, effectiveness is greatly dependent on the delivery format, targeted mental health problem and targeted purpose group. A study on digital mental health [36], synthesized evidence on digital health interventions targeting university students to evaluate their effectiveness. The results indicated that web-based online/computer delivered-interventions were effective or at least partially effective at decreasing depression, anxiety, stress and eating disorder symptoms. A controlled study [37] evaluated the feasibility of a simple "prototype" of a therapist-assisted computerized cognitive behavioral therapy (TacCBT) in young adult users affected by anxiety disorders. The preliminary results showed the benefits of the TacCBT program. Computerized cognitive behavioral therapy (cCBT) appeared to be a therapeutic strategy that is as effective as person-to-person CBT in the treatment of adults and young people with anxiety disorders.

So, within the future clinical implications of this study, it appears useful to emphasize the need for the implementation of relational and psychological interventions, even digital, to improve the mental health of vulnerable young subgroups during a global health emergency as the COVID-19 pandemic. The implementation of relational and psychological online services in academic institutions to contain, the evolution of psychopathological profiles among vulnerable young groups represents a fundamental challenge for the future. Indeed, the UN Agenda 2030 for Sustainable Development recomends to achieve the best possible level of health and wellbeing for all people living in the European Region (global goal 3) and recognizes that health and wellbeing must go hand-in-hand with strategies that improve the quality of education (global goal 4).

The strenght of the study is to gain a better understanding of the experiences and behaviors of young students during the global pandemic, investigating the impact of their changed habits on mental health and wellbeing. These findings contribute in several ways to the understanding of the effect of the restricions and provide a basis for the affirmation of the need for relational and psychological interventions. It's possible that health behavior was affected not just by public health restrictions, but also having had family members or friends who died or were severely sick due to COVID, as well as exposure to severe illnesses and rising mortality in the news. There are some major limitations in this study that could be addressed in future research. First, the effect estimates are based on observational studies, this makes the generalization of the results something to do with caution; second, the sample does not correspond to a precise sample design, but was built by drawing on institutional lists of students enrolled in degree courses with characteristics. This characteristic of the sample makes the impact of the research results on the knowledge of the problems less strong. The low participation rates are also major limitations.

## Supporting information

**S1 Table. Dietary habits of the respondents compared by participant cohort.** Percentages (Modal values in bold) or absolute values.
(DOCX)

**S2 Table. Physical activity practices of the respondents compared by participant cohort (IFIS).** Percentages (Modal values in bold) or absolute values.
(DOCX)

## Author Contributions

**Conceptualization:** Mario Corsi, Rémy Hurdiel, Philippe Masson.

**Data curation:** Thierry Pezé, Philippe Masson.

**Formal analysis:** Ivana Matteucci, Thierry Pezé.

**Funding acquisition:** Ivana Matteucci.

**Investigation:** Ivana Matteucci, Rémy Hurdiel, Alessandro Porrovecchio.

**Methodology:** Alessandro Porrovecchio.

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
