## [Decision Letter · Decision Letter 0]

3 Apr 2023

PGPH-D-23-00020

Health-related behavioral changes during the COVID-19 pandemic. A comparison between cohorts of French and Italian university students

Dear Dr. Matteucci,

Thank you for submitting your manuscript to PLOS Global Public Health. After careful consideration, we feel that it has merit but does not fully meet PLOS Global Public Health’s publication criteria as it currently stands. Therefore, we invite you to submit a revised version of the manuscript that addresses the points raised during the review process.

The reviewers want more information about methodology of the study. In your revision, please supply that.

We look forward to receiving your revised manuscript.

Kind regards,

Abram L. Wagner, PhD, MPH

Academic Editor

Journal Requirements:

Additional Editor Comments (if provided):

Reviewers' comments:

Reviewer's Responses to Questions

**Comments to the Author**

1. Does this manuscript meet PLOS Global Public Health’s publication criteria? Is the manuscript technically sound, and do the data support the conclusions? The manuscript must describe methodologically and ethically rigorous research with conclusions that are appropriately drawn based on the data presented.

Reviewer #1: Partly

Reviewer #2: No

2. Has the statistical analysis been performed appropriately and rigorously?

Reviewer #1: No

Reviewer #2: Yes

3. Have the authors made all data underlying the findings in their manuscript fully available (please refer to the Data Availability Statement at the start of the manuscript PDF file)?

Reviewer #1: No

Reviewer #2: Yes

4. Is the manuscript presented in an intelligible fashion and written in standard English?

Reviewer #1: Yes

Reviewer #2: Yes

5. Review Comments to the Author

Reviewer #1: The current study aimed to investigate the impact of the COVID-19 pandemic on choices and lifestyle changes in a sample of Italian and French university students.

The paper provides a modest scientific contribution to the wide panorama of studies investigating the general population's health and quality of life during the Covid-19 pandemic.

The topic is interesting enough, but significant changes and explanations should be made to improve the quality of the manuscript, especially in conceptual terms, for his acceptance.

Abstract section

In the ‘abstract section’, the Authors should briefly describe the survey period, the procedures, and the methodology used to recruit, assess, and analyze the considered variables. The results are summarized without any statistical data, and there are no conclusions concerning future clinical-social implications. The Authors should add them. The keyword "mental health" could be entered.

Introduction section

The bibliography in the ‘introduction section is appropriate and updated but very poor. However, it would be useful to mention, for example, recent international studies (1, 2, 3, 4, 5, listed below) on the impact of Covid -19 pandemic on college students.

Some authors focused on psychological, emotional, and cognitive correlates of social confinement in a sample of college students, integrating qualitative and quantitative analyses to identify potential predictors of traumatic distress during Covid -19 and low academic performance during Covid -19. The Authors identified the thinking style “all or nothing” as the strongest predictor of traumatic distress in a sample of university students. Another recent UK population survey showed young adults coping poorly, and pre-existing mental health conditions were significantly associated with suicidal thoughts and self-harm.

An Australian study showed that loneliness moderated the mediating effect of a resilient mindset in the coronavirus stress and depressive symptoms association among college students. The authors do not explain their study's objectives and/or hypotheses at the end of the introduction.

Materials and methods section

The methods and procedures are well described. The tables are exhaustive for data presentation. The Authors should clarify how focus groups were organized to develop the survey (questions/items). It is not clear who conducted this investigation. Is this perhaps, for example, a sort of Listening and Counselling Service for college students? The Authors should make this clear.

Discussion and Conclusions section

The discussion is incomplete: there are no comments on the results with comparisons with other surveys conducted during the pandemic, strengths and limitations of the study, study conclusions, and clinical/social future implications. Authors should make the suggested enrichments.

The authors make no mention of the University Listening and Consultation Services, a well-represented reality in the academic world, to help students cope with the emotional distress related to academic difficulties.

Within the future clinical implications, it would be useful to emphasize the need for the implementation of psychological interventions, even digital, to improve the mental health of vulnerable young subgroups during a global health emergency as the Covid-19 pandemic, and contain, as far as possible, the evolution of psychopathological profiles represents a fundamental challenge (6, 7 listed below)

Overall, the manuscript's clarity is good enough in the language style, syntax, and sentence construction.

In the ‘Conclusions section,’ the Authors should report how their study contributes to achieving the Global Goals of the 2030 Agenda for Sustainable Development.

1. https://pubmed.ncbi.nlm.nih.gov/33384623/

2. https://pubmed.ncbi.nlm.nih.gov/36377653/

3. https://pubmed.ncbi.nlm.nih.gov/36339876/

4. https://www.ncbi.nlm.nih.gov/pmc/articles/PMC8718341/

5. https://pubmed.ncbi.nlm.nih.gov/34526153/

6. https://www.ncbi.nlm.nih.gov/pmc/articles/PMC7005461/

7. https://pubmed.ncbi.nlm.nih.gov/35382010/

Reviewer #2: Thankyou for giving me an opportunity to review this. The title of the study is quite attractive but the findings are very limited and not well explained. On the other hand, the methodology is an important part of any study and in this study, methodology is not cleared. There is no sampling techniques, no rules for sample size calculation and sample size is not justified. Also there is no clarity about participation rate. Most importantly, discussion part is not satisfied and not elaborate all the important points of this study. I appreciate your efforts. But improvement is needed. Thankyou.

6. PLOS authors have the option to publish the peer review history of their article (what does this mean?). If published, this will include your full peer review and any attached files.

**Do you want your identity to be public for this peer review?** For information about this choice, including consent withdrawal, please see our Privacy Policy.

Reviewer #1: **Yes: **Rita Roncone

Reviewer #2: No

---

## [Editor Report · Decision Letter 1]

10 Jul 2023

PGPH-D-23-00020R1

Health-related behavioral changes during the COVID-19 pandemic. A comparison between cohorts of French and Italian university students

Dear Dr. Matteucci,

Thank you for submitting your manuscript to PLOS Global Public Health. After careful consideration, we feel that it has merit but does not fully meet PLOS Global Public Health’s publication criteria as it currently stands. Therefore, we invite you to submit a revised version of the manuscript that addresses the points raised during the review process. My comments and feedback are listed below.

A rebuttal letter that responds to each point raised by the editor. You should upload this letter as a separate file labeled 'Response to Reviewers'.A marked-up copy of your manuscript that highlights changes made to the original version. You should upload this as a separate file labeled 'Revised Manuscript with Track Changes'.An unmarked version of your revised paper without tracked changes. You should upload this as a separate file labeled 'Manuscript'.

We look forward to receiving your revised manuscript.

Kind regards,

Sanghyuk S Shin

Academic Editor

Journal Requirements:

Additional Editor Comments (if provided):

Thank you for your attention to the comments raised by prior reviewers. I believe most of them have been addressed adequately. However, there are remaining issues that should be addressed for the article to meet the journal’s publication criteria for rigor.

- Given that the study used a cross-sectional study design and the analysis involves descriptive statistics, the causal claims throughout the manuscript are not supported by adequate evidence. Please go through the entire manuscript and delete or modify the text so that no causal inference is made. For example, on Page 3, you state: Page 3 “Italian students suffered more than French students in their well-being due to the harder social restrictions applied in Italy”. However, your analysis does not specifically isolate “social restrictions” as a causal factor independent of confounders that might explain the different results between the two countries. These types of causal conclusions are found in several places in the manuscript.

- In the Methods, you state that ethical review is not required for human participants. However, research ethics boards are specifically designed to provide oversight on human subjects research. Please explain.

- In the Abstract, you use the term “prevalently” to describe the study design. I believe this is not a standard term. Perhaps you mean “cross-sectional”? Please change to clarify the meaning of this term/sentence.

- The Methods section has been greatly improved in this version. However, it is still unclear how the students were selected for recruitment. Were all students in the directory contacted via email? How many students were attempted? What proportion were contacted? How many refused after contact?

- Please make sure your manuscript follows the STROBE guideline for cross-sectional observation studies: https://www.strobe-statement.org/checklists/

- Please discuss the following points in the limitation section

o It’s possible that health behavior was affected not just by public health restrictions, but also having had family members or friends who died or were severely sick due to COVID. And also exposure to severe illnesses and rising mortality in the news. It seems that your analysis does not distinguish these and other factors that could affect health behavior outcomes. Please discuss this limitation.

o The low participation rates are major limitations. Please include a discussion of this.
---

## [Editor Report · Decision Letter 2]

25 Jul 2023

Health-related behavioral changes during the COVID-19 pandemic. A comparison between cohorts of French and Italian university students

PGPH-D-23-00020R2

Dear Associate professor Matteucci,

We are pleased to inform you that your manuscript 'Health-related behavioral changes during the COVID-19 pandemic. A comparison between cohorts of French and Italian university students' has been provisionally accepted for publication in PLOS Global Public Health.

Best regards,

Sanghyuk S Shin

Academic Editor